# E-Taste: Taste Sensations and Flavors Based on Tongue’s Electrical and Thermal Stimulation

**DOI:** 10.3390/s22134976

**Published:** 2022-06-30

**Authors:** Asif Ullah, Yifan Liu, You Wang, Han Gao, Hengyang Wang, Jin Zhang, Guang Li

**Affiliations:** 1State Key Laboratory of Industrial Control Technology, College of Control Science and Engineering, Zhejiang University, Hangzhou 310027, China; asifkh@zju.edu.cn (A.U.); 22032135@zju.edu.cn (Y.L.); gao_han@zju.edu.cn (H.G.); 11432014@zju.edu.cn (H.W.); guangli@zju.edu.cn (G.L.); 2School of Computer and Communication Engineering, Changsha University of Science and Technology, Changsha 410114, China; mail_zhangjin@163.com

**Keywords:** E-taste, taste stimulation, digital flavors, human-computer interface, virtual reality

## Abstract

Taste is a key sense that helps identify different food types and most of this work was carried out on primary tastes rather than generating different flavors. In this work, we proposed a plan to create other flavors rather than primary tastes, adjusted the electrical (40–180 µA) and thermal stimulation (20–38 °C and 38–20 °C), and revealed the digital coding for multi-flavors. Our results showed that different combinations of digital coding could generate different flavors and that tastes related to different stimuli are easy to develop. The novelty of this work is to design other types of flavors and primary tastes. The experimental results demonstrated that the novel method proposed for digital taste coding could realize primary tastes (sweet, sour, salty, spicy, and mint) and mixed flavors. Furthermore, some innovative sensations have been realized, which are sprite, soda water, sweet-sour, salty-sweet, and salty-mint sensations. We presume that this innovation could digitally enhance various flavors.

## 1. Introduction

Several researchers have worked on five primary basic senses: touch, smell, hearing, taste, and vision over the past few years. Among the five primary senses, taste is one of the five basic senses to identify the sensation of food. The sense of taste is vital in stimulating human beings to devour food variations [1]. Previous research indicates that when people eat their favorite foods, it causes the release of β-endorphins, which is a component that helps improve mood [1]. According to the consumer survey report [2], taste is the primary influence on food selection. Although behavioral, sociocultural, and economic factors influence dietary trends [3,4], consumers report that their diet choices are most often affected by how foods taste [2]. The chemical senses of taste, olfaction, and the oral perception of texture are included in the general concept of a food’s “taste” [5,6,7]. There is an indication that the obese are overly sensitive to the pleasant or hedonic aspects of food, favoring spicy or flavored foods [8]. Prior researchers investigated the taste preferences of sweetness in chocolate milk, yogurt, and black tea [9] and, in addition, the food matrices’ ability to improve the salty taste with artificial flavors [10]. Sensory perceptions and preferences influence food preferences and eating habits for food taste, aroma, and consistency [8,11]. Sugar and fat-containing foods are universally preferred [8,11], whereas bitter tastes are universally disliked [12]. As a result, taste is essential to the soul as food is to the body [1].

Taste is frequently confused with flavor, which is the combined tactile involvement of olfaction and gustation. Gustatory signals originate in the oral cavity taste buds’ tactile conclusion organs. They are activated by the interaction of water-soluble compounds with the apical tips of the epithelial cells of taste buds. Neurons produce olfactory signals in a specialized fix of nasal epithelium activated by unstable compounds. Even though the sensory organs for taste and smell are particular, their signals are coordinated within the orbitofrontal and other zones of the cerebral cortex to create flavors and intervene in food recognition [13]. Nevertheless, some researchers have worked to control the harshness of tobacco by developing e-cigarettes with ice-hybrid flavors that contain non-methanol synthetic cooling agents to produce fruity or sweet properties [14].

The electronic tongue is a cutting-edge technology designed for assessing and analyzing food flavors such as umami taste [15,16]. In this context, umami-based tastes were also distinguished from basic flavors using nanoparticle-based electronic tongues [17]. It has the capability to evaluate product quality [18]. Furthermore, Kumar et al. identified the response of different samples of black tea by introducing an electronic tongue [19]. The authors [20] initiated a remote electronic tongue system to identify honey from different botanical origins.

During recent times, very few researchers have worked on a digital taste sensation. BeanCounter is a network monitoring and computer application that assigns jellybeans based on their memory and data communication sensitivities [21]. TasteScreen allows users to lick their screens (which contain a mixture of flavor cartridges and chemical flavors) to taste nourishment on their computer screens [21]. Other research facilities used practical magnetic resonance imaging to investigate higher-order cortical processing to address the interaction between taste innovation, disposition, and appetite control [22]. Furthermore, a few endeavors were related to electrical stimulation of the tongue in the medical area [23,24,25]. Cruz et al. [26] presented thermal stimulation of the tongue, demonstrating that warming and cooling small tongue areas can produce taste sensations. Lawless et al. [27] proposed metallic taste based on electrical and chemical stimulation to investigate the similarities and differences between metallic-based taste, electrical stimuli, and divalent salt provisions with ferrous sulfate. They revealed that a weak electric pulse on the tongue produces a metallic-like taste sensation. According to the author [28], certain tastes, including salty and sour, could be related to the ambient temperature and chemical transcriptional channels. Those partakers who experienced taste perceptions due to cooling or heating have been listed as the “thermal tasters” (TT) [29].

Concerning virtual reality (VR), nobody has created a constrained virtual environment to evoke various taste senses. As mentioned in [30,31,32], some researchers innovated sweet taste equipment. Prior research has focused on developing a sweet flavor (or any former primary taste). Other researchers such as Karunanayaka et al. created an interface that warms the tongue to produce sweetness [33]. They went on to say that swiftly warming the tongue has a sweet and fatty sensation, whereas cooling it produces a minty sensation and pleasantness. Nimesha et al. [34] produced excellent work on digital taste. Using electrical and thermal stimuli, they created four basic tastes (sweet, sour, bitter, and salty) on the tongue’s surface.

Some researchers stimulate the tongue by heating and cooling the tongue, which generates sweet and sour sensations [35]. Furthermore, PROP taster status (PTS) and thermal taster status (TTS) for oral stimuli have been introduced [36]. They revealed that heating or cooling different tongue portions could induce different tastes. Cheok et al. [30] proposed a device that could provoke sweet sensations via thermal stimuli. In the meantime, the creators addressed a layout that might improve the perception of switching the food by adjusting warm sensations to the nose’s skin, causing changes in the skin’s temperature related to delightful or repulsive emotions [37]. Nevertheless, except for primary tastes, the outcomes of this research were not prolonged to many further flavors. As a result, determining the optimal electrifying constraints to create diverse flavors remains a mystery, and e-taste (electronic taste) interfacing with multisensory or virtual reality applications is mainly unexplored.

The majority of the researchers mentioned above focused on the primary tastes. The main challenge is generating different digital flavors based on basic digital tastes. To widen the scope of digital taste research, we proposed a system that can generate different flavors on the tongue by combining other basic tastes (sweet, sour, bitter, and salty). Various stimuli are applied to the tip of the human tongue by changing current and temperature via silver electrodes in the control framework.

The novelty of this work is to generate different mixed sensations to get other types of flavors digitally based on hybrid (electrical and thermal) stimulations on the tip of the tongue that could be implemented in various fields such as medicine, virtual reality games, and many other aspects of life. Moreover, the current research may consider future avenues that could stimulate the tongue of a hypogeusiatic person (taste disorder) to generate various tastes.

## 2. System Description

The proposed system consists of three submodules: command center, control system, and tongue interface, as shown in Figure 1. The command center (computer) instructs the control system to process the output current and temperature. An Arduino wire connects the command center to the control system. The control system (Figure 2a) comprises two parts: electrical and thermal. Seven wires connect the tongue interface to the control system (two wires for silver electrode plates, two wires for the Peltier module, and three wires for the LM35 temperature sensor). The tongue interface (Figure 2b) consists of two silver electrodes (50 mm × 15 mm × 0.2 mm), a Peltier module (40 mm × 20 mm), a temperature sensor (LM35), and a larger heatsink to cool down the Peltier module to the desired temperature. The main advantage of using silver electrodes is that they are good conductors of electricity and temperature.

Furthermore, when compared to other electrodes, silver does not cause metallic sensations in humans. A thermal compound paste between the silver electrodes and the Peltier module has been used. The thermal compound paste is a polymeric fluid composite with significant electrically insulating fractions but a high thermal conductivity pitch. The main reason for using this compound is to maximize heat transfer and dissipation by eliminating air gaps or spaces (insulators) from the interface area. The main advantage of using a bigger heatsink is the quicker cooling down of the Peltier module.

During electrical stimulation, a digital potentiometer (MCP41010) with a constant current source was provided to all the subjects, as shown in the schematic diagram (Figure 3). The main reason for using a digital potentiometer is that Arduino can easily control it. The potentiometer has 256 taps with resistances of 10 kΩ, 50 kΩ, and 100 kΩ. The taps of the potentiometer were controlled through the Arduino to control the current magnitude. The 256 taps were set based on the contact resistance connected to the silver electrodes. We made a table of every possible tap to give a specific magnitude of the current. For example, if we set the taps from 55 to 225, it provides a current extent from 40 µA to 180 µA across the contact resistor connected to the silver electrodes. A constant current source is used to provide constant current no matter what the tongue’s resistance is (each person’s tongue’s resistance varies). During the experiment, a square wave pulse (both AC and DC sensation) with a unique current ranging from 40 µA to 180 µA was applied through silver electrodes to the tip of the tongue. Previous research has determined that the frequency range for stimulating the tongue is 50–1000 Hz [34]. As a result, 490 Hz was chosen as the average frequency. During thermal stimulation, a PWM motor driver was employed to heat the Peltier module connected to the silver electrodes up to 38 °C (AO1 on/AO2 off). When the temperature sensor (LM35) detects a temperature of up to 38 °C, it sends a command to the control system to change the current path, allowing the Peltier module to cool down to 20 °C with the help of a heatsink (AO2 on, AO1 off) as highlighted in Figure 3.

## 3. Experimental Procedure and Results

The study was conducted with 23 participants (fifteen men and eight women) between the ages of 23 and 45 (±SD, 6.72586). All the participants were non-smokers and had no taste disorders. The experiment was approved by the Ethical Committee of the College of Biomedical Engineering and Instrument Science, Zhejiang University (Zheda Shengyi Huishen 2021 No. 14).

The participants were advised not to eat harmful foods or drink alcohol 3 h before the experiment. The experiment was designed according to the following steps:The current and temperature stimulations of the e-device were checked first;The participants were given the table of the hedonic scale, as shown in Table 1;The participants were recommended to sit comfortably on the chair, hold the tongue interface, and place the silver electrodes on the tip of the human tongue;In the next step, the current (40–180 µA), temperature (20–38 °C and 38–20 °C), and hybrid (electrical and thermal) stimulations were given to the tip of the human tongue through silver electrodes, and the participants were asked to share their taste sensations. The stimulation for different taste sensations varies, as shown in Table 2;After each stimulation, the participants were asked to scale the taste sensations according to the hedonic scale;Between each stimulation, the participants took a break of 5 min depending on the participant’s tongue’s ability to become normal.

The experiment was conducted in two different ways. Firstly, the investigation was conducted on the participants without telling them about the taste stimulants. They were instructed to communicate their feelings when stimulating the tongue, proving the device was working and generating different tastes and flavors. Secondly, the participants were given a list of different tastes and flavors, stimulated the tongue with random tastes, and were asked to identify the taste. Some participants could not distinguish between different flavors participating in the experiment. However, when they were given a list of various tastes and flavors, most of them could specify the exact flavor.

The experiment was conducted with three types of stimulants: electrical, thermal, and hybrid (electrical and thermal) stimulations. Electrical stimulation was applied to the tip of the tongue through silver electrodes with a varying magnitude of current (40–180 µA). In contrast, thermal stimulation was applied to the tongue with varying temperatures of 20–38 °C and 38–20 °C, while hybrid stimulation with a prearranged magnitude of current (40–180 µA) and varying temperatures (20–38 °C and 38–20 °C) was given to the tip of the tongue to generate different types of sensations.

Table 2 shows different taste sensations and the magnitude of the other taste sensations. During the experiment, we observed that sour, salty, and bitter are mostly related to electrical stimulation. Several researchers described bitterness as related to the lateral part of the tongue [38]. We applied the range of current (60–140 µA) to both the frontal and lateral parts of the tongue, and we found that the bitterness on the lateral part of the tongue is more assertive than on the frontal part. We also discovered that the strength of salty and sour increases with the current magnitude.

During thermal stimulation, the strength of the spicy feeling increases with the temperature rise, while the power of the minty sensation increases with the decrease of temperature, as shown in Figure 4. However, some participants reported slightly salty, moderately bitter, and sour feelings. Moreover, we found that sweet sensation is mainly related to thermal stimulation (heating up and cooling down the tongue) according to a specific magnitude of temperature. Some participants realized the sweet sensation while heating the Peltier module connected to the silver electrodes and placed on the tip of the tongue, and some felt it while cooling down.

Our focus was to develop new flavors (Figure 5); therefore, we applied hybrid (electrical and thermal) stimulations to the tip of the tongue via silver electrodes and mixed different taste sensation ranges. Several participants reported different kinds of flavors and mixed sensations during hybrid stimulation. Some participants noted a sprite and soda water sensation when increasing the current magnitude from 60 to 80 µA and decreasing the temperature from 30 to 20 °C. Combining the taste sensation’s range of salty (electrical stimulation) and sweet (thermal stimulation) gives salty-sweet feelings. Moreover, applying a current magnitude of 60–180 µA with varying temperatures from 20–38 °C and 38–20 °C gives a sweet-sour sensation. During hybrid stimulation, salty-mint sensations were easy to generate. Increasing the current magnitude gives a salty sense, while at the same time, decreasing the temperature magnitude gives you a minty feeling. The current range of sour (60–180 µA) and bitter sensations (60–140 µA) are almost the same; while stimulating the tongue, the participants felt both sour and bitter sensations, and some of them could not differentiate their feelings. Table 3 shows the mean value of taste sensations and the standard error of the mean (SEM) for 23 participants.

This study also reveals that the taste lasts longer by stimulating the tongue for a longer period. For sprite/soda water sensation, after removing the electrodes from the tip of the tongue, the participants can feel the sensation for up to 10 s. For salty sensations (40–70 µA), the taste lasts on the tongue for 2 s. Likewise, for other taste sensations, the taste lasts on the tongue for 4–8 s, depending on the taste stimuli and participant. Some participants were sensitive to taste and could strongly feel different flavors and taste sensations, as shown in Figure 6. Based on a scale of 1, some participants can feel a value of up to 0.9 of mean taste sensation.

Additionally, single-factor ANOVA was adopted to study the e-taste further, as shown in Table 4, which shows that the F-statistic value is less than the F-critical value and proves that the test is statistically significant. The experiments were repeated 3 times with 23 participants through various stimuli, as shown in Figure 7. Experimental results show a minimal significant difference among the different trials. The error bar depicts a 90–98% confidence interval (CI).

## 4. Discussion and Future Work

Our ‘e-taste device’ revealed strong and modified taste sensations for sweet, sour, salty, mint, spicy, bitter, and various flavors like sprite, soda water, sweet-sour, salty-sweet, and salty-mint sensations. The experiment showed that the e-taste device was able to generate various flavors and primary tastes. This study also demonstrated that varying the current magnitude and heating and cooling the tongue generates different flavors. To the best of our knowledge, this is the first time different flavors added to the primary tastes have been introduced. The impact of e-taste could be used to enhance various flavors in multidisciplinary areas. This study also describes that sweetness, spiciness, and minty sensations are related to thermal stimulation, while sour, bitter, and salty are mostly related to electrical stimulation. The e-taste device can be used to make virtual food and beverages in various aspects of life. The comparison of related works and e-taste device are illustrated in Table 5. Our device enhanced various sensations that were not previously reported. One of our future goals is to seek data in various sensory stimuli to deliver further flavors.

We are also engaged in combining the e-taste with other stimuli, such as various types of visual, auditory, touch, and olfactory stimuli. We would like to dedicate future research to improving the silver plate assembly, heatsink design, and Peltier device with varying speeds and peaks. Furthermore, we still need to clarify some constraints to make this device easily accessible for everyday life. The device’s lack of user-friendliness for frequent usage is its major flaw. Some users are cautious about placing the silver electrode plates on the tongue, as they are concerned about issues such as burning and sanitation. As a result, finding more innovative ways to convey taste sensations would be beneficial, as would collaborating with researchers from various disciplines, such as nutrition, flavor, and pharmaceutics, to improve the technology.

After acknowledging the limitations discussed earlier in this section, we hope that e-taste will be useful in future daily life. Sensing and recreating taste and smell experiences, on the other hand, remains a significant challenge. After more evaluation and improvements to our device, we presume we will be able to propose stimulation parameters for various flavors. E-taste could also be combined with olfactory actuation techniques to develop more complicated and relevant flavor perceptions.

## 5. Conclusions

Most of the previous systems were based on generating five basic tastes. Few attempts have been made to create different flavors based on primary tastes. The proposed approach was able to create different flavors along with the basic tastes. Experimental results showed that mixing primary tastes could develop other flavors. The most innovative flavors revealed in this paper are sprite, sweet-sour, salty-sweet, salty-mint, and various other mixed sensations. We also found that young participants could sense different flavors of sensations compared to aged participants. It may be due to the difference in taste buds among various participants, and the quality of the taste sensation taste dropped with age. In addition, the system could be used in VR games shortly, as other senses are being introduced in the VR world. This field’s applications could create more opportunities in daily activities such as medicine, gaming, and the human-computer interface (HCI). Further investigation is needed regarding taste sensations and different types of flavors.

## Figures and Tables

**Figure 1 sensors-22-04976-f001:**
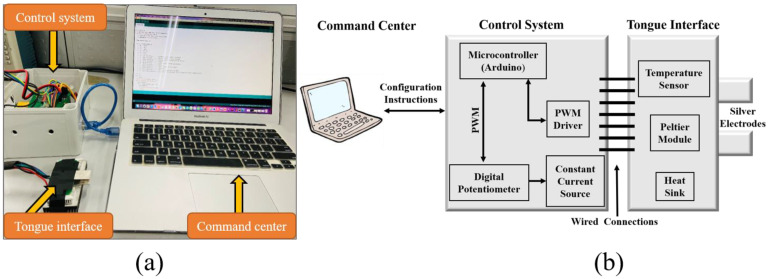
Digital taste system (**a**) implementation and (**b**) system architecture.

**Figure 2 sensors-22-04976-f002:**
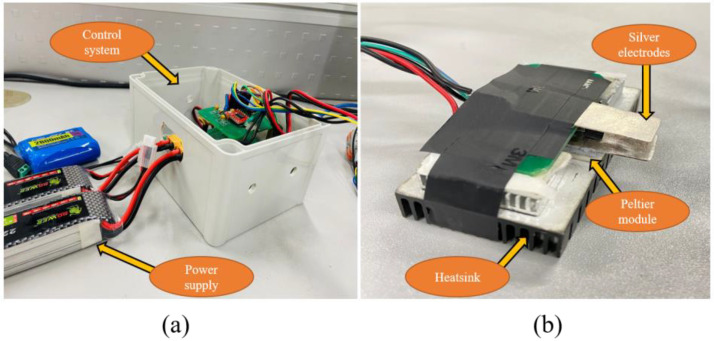
Digital taste generating system (**a**) control system (**b**) tongue interface.

**Figure 3 sensors-22-04976-f003:**
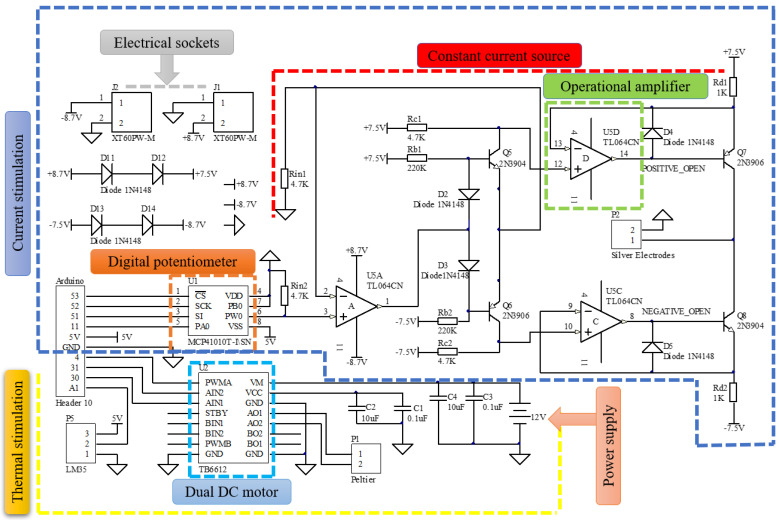
Schematic diagram of e-taste.

**Figure 4 sensors-22-04976-f004:**
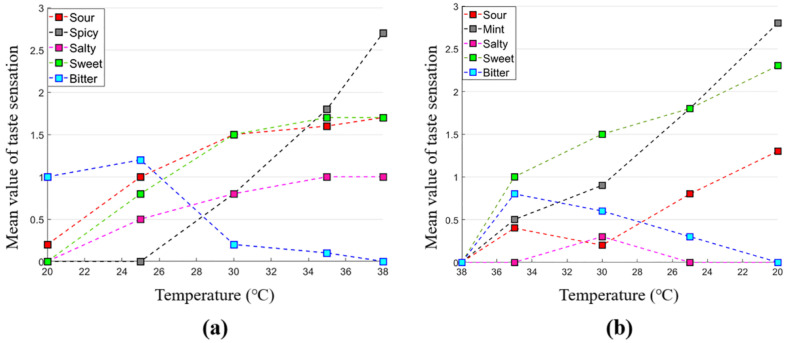
Thermal stimulation (**a**) heating up to 38 °C (**b**) cooling down up to 30 °C.

**Figure 5 sensors-22-04976-f005:**
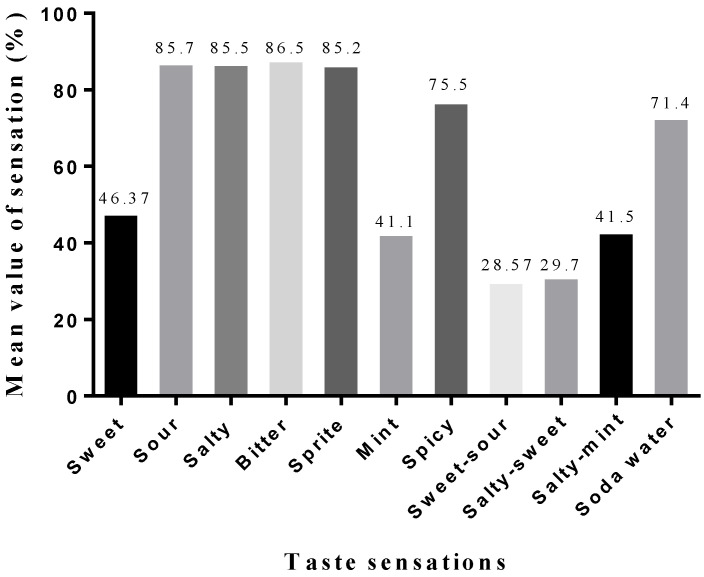
Different taste sensations and flavors versus mean value of sensation (%).

**Figure 6 sensors-22-04976-f006:**
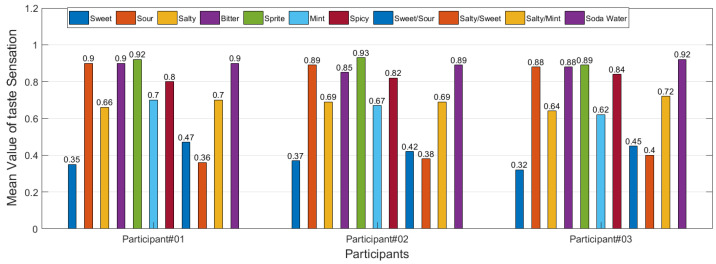
Mean value of taste sensations of 3 participants with highly sensitive taste buds.

**Figure 7 sensors-22-04976-f007:**
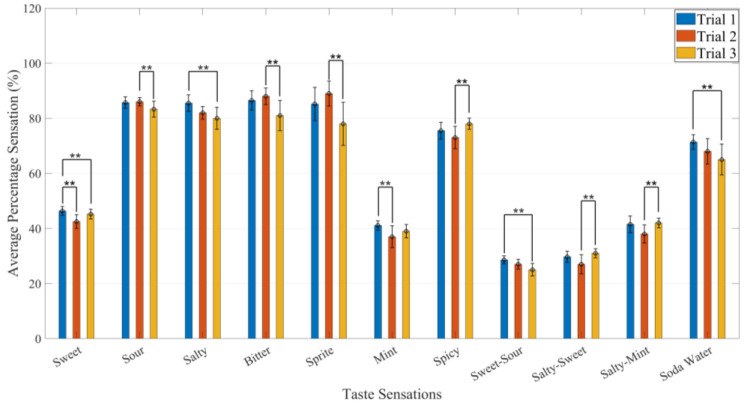
The average percentage of taste sensations of 23 participants with repeated experiments. Error bar depicts 90–98% CI. All data are expressed as mean ± SD ^⁎⁎^ *p* ≤ 0.05.

**Table 1 sensors-22-04976-t001:** Taste sensations scale.

0				1		
−3	−2	−1	0	1	2	3
Extreme dislike	Moderate dislike	Slight dislike	No sensation	Slight sensation	Moderate sensation	Extreme sensation

**Table 2 sensors-22-04976-t002:** The magnitude of taste sensations.

Taste Sensations	Magnitude/Range of Sensation
Sweet	25–35 °C and 35–25 °C
Sour	60–180 µA or 20–30 °C
Salty	40–70 µA
Sprite/soda water	60–180 µA and 30–20 °C
Bitter	60–140 µA
Mint	Below 25 °C
Spicy	Above 33 °C
Sweet-sour	60–180 µA and 25–35 °C and 35–25 °C
Salty-sweet	40–70 µA and 25–35 °C and 35–25 °C
Salty-mint	40–70 µA and below 25 °C

**Table 3 sensors-22-04976-t003:** The mean value of taste sensations and standard error of the mean of 23 participants.

Taste Sensations	Mean Value ± Standard Error of the Mean
Sweet	0.463 ± 0.1
Sour	0.857± 0.07
Salty	0.855 ± 0.07
Bitter	0.865 ± 0.07
Sprite	0.852 ± 0.07
Mint	0.411 ± 0.1
Spicy	0.755 ± 0.08
Sweet-sour	0.285 ± 0.09
Salty-sweet	0.297 ± 0.09
Salty-mint	0.415 ± 0.1
Soda water	0.714 ± 0.09

**Table 4 sensors-22-04976-t004:** Single-factor ANOVA taste.

Source of Variation (SV)	Sum of Squares (SS)	Degree of Freedom (df)	Mean Square (MS)	F_statistics_	*p*-Value	F_crit_
Between groups	13.17	10	1.32	7.2	6.2 × 10^−10^	1.87
Within groups	44.28	242	0.18			
Total	57.45	252				

**Table 5 sensors-22-04976-t005:** Related works versus e-taste devices.

Title of the Research Paper	Senses Generated	Revised Sensation	Excitation Constraints	Reference
Metallic Taste from Electrical and Chemical Stimulation	- Metallic-based taste, electrical stimuli, and divalent salt provisions with ferrous sulfate	Nil	Nil	[27]
Thermal stimulation of taste	- Demonstrate that warming and cooling small tongue areas can produce taste sensations	Nil	- Heating (20–35 °C) - Cooling (≤20 °C) - Rate ±1.5 °C s^−1^	[26]
Thermal Taste Actuation Technology	- Swiftly warming the tongue has sweet and fatty sensations, whereas cooling produces a minty sensation and pleasantness.	Significant improvements in the sweetness of sucrose-based sweet solutions were achieved.	- Heating (25–40 °C) Cooling (25–10 °C) - Heating rate (1.5 °C s^−^^1^, 1 °C s^−^^1^, 0.66 °C s^−^^1^) - Cooling rate (0.5 °C s^−^^1^)	[33]
Digital taste: electronic stimulation of taste sensations	- Generated basic taste sensations (sweet, sour, bitter, and salty) on the tongue’s surface.	Nil	- Heating (20–35 °C) - Cooling (≤20 °C) - Heating rate (0.33 °C s^−1^) - Cooling rate (0.28 °C s^−1^)	[34]
Digital Taste and Smell Communication	- Cooling produced sourness	Nil	Cooling (35–20 °C)	[32]
Virtual sweet: Simulating sweet sensation using thermal stimulation on the tip of the tongue	- Produced Sweetness	Nil	Heating to Cooling (20–35 °C and 35–20 °C) Cooling to Heating (35–20 °C and 20–35 °C)	[31]
E-taste: Taste sensations and flavors based on tongue’s electrical and thermal stimulation (this paper)	- Generate significant effect for sweet, sour, bitter, salty, mint, and spicy sensations - Generate different flavors (Sprite, Soda water, sweet-sour, salty-sweet, and salty-mint)	Generate various types of flavors by combining different taste sensations	- Thermal stimulation: Heating (20 °C – ≥38 °C) - Cooling (38 °C – ≤20 °C) - Electric stimulation: (40–180 µA)-Hybrid stimulation includes both electrical and thermal stimulation - Heating rate (0.65 °C s^−1^) - Cooling rate (0.5 °C s^−1^) - Current rate: (10 µAs^−1^)	Current work

## Data Availability

Data about personal sensory evaluation scores are not available due to contract restrictions.

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
