# Peer review of "E-Taste: Taste Sensations and Flavors Based on Tongue’s Electrical and Thermal Stimulation"

_sensors, 2022, doi:10.3390/s22134976_

Round 1

Reviewer 1 Report

The authors claim to develop complex digital flavors based on basic digital taste. The novelty of the work is to use digital taste coding to generate mixed flavor sensation.

Generally, the methodology is unclear and described badly, moreover there seem to be fundamental issues with the design of experiments and analysis of the results.

Comments:

1.     The 11th line of Introduction “There is also a pleasure…” is incomplete

2.     On the 1st line of introduction author mentioned both sight and vision as primary basic senses even though they both mean the same.

3.     In the 2nd paragraph of the introduction (line 23) the author’s wrote “… their signals are coordinates..”. Should be written as “are coordinated”.

4.     On page 2, line 12 the preposition “in” is unnecessary.

5.     On page 3 under system description these is an unnecessary use of “)” punctuation (line 13)

6.     On Page 3, figure 1 should be re-scaled to fit the page.

7.     In system description: when the authors mention tip of the tongue do they mean the tip of the silver electrodes or actual human subject? The authors switch back and forth between silver electrodes and tip of the tongue without a clear distinction making it difficult to follow through.

8.     On page 4, 4th paragraph the author wrote “we use a PWM motor driver to drive current in two ways as mentioned in figure 3”. Figure 3 doesn’t reciprocate this statement.

9.     Figure 3 needs to be labelled properly.

10.  On page 5, the authors mentioned the step 5 of their process : “….the participants were asked to scale the sensation according to hedonic scale” The spectrum of this scale is based on like and dislike towards the sensation. But the goal of this paper seems to generate mixed flavors. How does this step co-relate with the goal?

11.  On page 5, paragraph 2 the line “ secondly, the participants …. Were asked to imagine the taste and give commands through a computer to stimulate the tip of the tongue” This statement is confusing. On the following paragraph the authors mentioned the parameters used for the experiments. If, the parameters given were already able to generate the mixture of flavors what is the purpose of this step? How does it help with the goal?

12.  On page 5, paragraph 5, the authors talk about how heating or cooling the tongue affects the taste sensation. This is very confusing. Are they heating the electrodes and then holding it to the tongue?

13.  On page 5, paragraph 5 the authors wrote “Moreover, we found that sweet is mainly related to heating up and cooling the tongue according to a specific magnitude of temperature” This statement is confusing.

14.  On page 7 figure 5, the authors showed the taste sensation percentage across the 23 participants. The percentages for primary tastes are much higher compared to the mixed flavors (sweet-sour, salty-sweet, etc.). In this case can the author claim that they were successful in generating these flavors accurately. If the participant’s answers were collected based on the hedonic scale, then the figure just represent the participant’s like and dislike towards the tastes and not the actual flavors. There is a fundamental flaw in the design of the study.

15.  On page 8, under discussion and future work the author states “ The experiment showed that E-Taste device effectively generate various flavors rather than primary tastes.” This statement seems untrue.

16.  There are punctuation mistakes throughout the manuscript. For example, in abstract the author used hyphen (-) in between words. Example: On 4th line of the abstract the author wrote Pre-vious, on 8th sentence author wrote influ-ence. This use of hyphen is present throughout the manuscript.

17.  The degree symbol for temperature should be fixed throughout the manuscript.

Author Response

The authors response has been attached.

Reviewer 2 Report

Comments about the paper:   According ot the authors, "The novelty of this work is to generate different mixed sensations to get other type of flavors digitally based on hybrid (electrical and thermal) stimulations on the tip of the tongue that could implement in various fields such as medicine, virtual reality games and many other aspects of life." In addition "The study was conducted with 23 participants (fifteen males and eight females) between the age of 23 to 45 ". "The experiment was conducted with three types of stimulants: electrical, thermal, and hybrid (electrical and thermal) stimulations. Electrical stimulation was applied on the tip of the tongue with a varying magnitude of current (40-180μA). In contrast, thermal stimulation was applied on the tongue with varying temperatures from 20-38-20 oC, while hybrid stimulation with a prearranged magnitude of current (40-180μA) and varying temperature (20-38-20 oC) was given to the tip of the tongue through silver electrodes to generate different types of sensations." "Three times repeated experiments were done on 23 participants with various stimuli." It is a reasonable claim and acceptable procedure for publication.  There are still a few formatting errors (e.g. pre-sent, re-search, pro-vided, ap-proved). "lm35" should be written as "LM35", as in other parts of the text. Symbol for micro appears as "u" instead of the Greek letter. When citing an author, it should be avoided to write, for instance, "Kumar S et al.". Could the "S" not be omitted? Photographs of the system should not be in the introduction. What is the "scheme zero" from Table 3? The Authors included references to the discussion through Table 5. However, despite not easy to cite recent papers pertinents to this work, this reviewer suggests a few to further improve this manuscript: Leventhal, A.M., Tackett, A.P., Whitted, L., Jordt, S.E. and Jabba, S.V., 2022. Ice flavours and non-menthol synthetic cooling agents in e-cigarette products: a review. Tobacco Control.
Mohamed, S., Khaled, S., Dahshan, M., Mohamed, B., Badr, M. and Hussein, K., 2022, May. A Review of Electronic Tongue for Liquid Classification. In 2022 2nd International Mobile, Intelligent, and Ubiquitous Computing Conference (MIUCC) (pp. 310-315). IEEE.
Ross, C.F., 2021. Considerations of the use of the electronic tongue in sensory science. Current Opinion in Food Science, 40, pp.87-93.
Wang, K., Zhuang, H., Bing, F., Chen, D., Feng, T. and Xu, Z., 2021. Evaluation of eight kinds of flavor enhancer of umami taste by an electronic tongue. Food Science & Nutrition, 9(4), pp.2095-2104.
Hensel, R.C., Braunger, M.L., Oliveira, B., Shimizu, F.M., Oliveira Jr, O.N., Hillenkamp, M., Riul Jr, A. and Rodrigues, V., 2021. Controlled Incorporation of Silver Nanoparticles into Layer-by-Layer Polymer Films for Reusable Electronic Tongues. ACS Applied Nano Materials, 4(12), pp.14231-14240. E. M. Rodríguez, M. R. Cavallari, G. S. Braga, E. F. G. Rodríguez and F. J. Fonseca, "Basic tastes classification using thin-film transistors based on poly (3-hexylthiophene)," 28th Symposium on Microelectronics Technology and Devices (SBMicro 2013), 2013, pp. 1-3, doi: 10.1109/SBMicro.2013.6676117.

Author Response

The authors response has been attached.

Reviewer 3 Report

Dear Authors,

I have been carefully reviewing your manuscript entitled “E-Taste: Taste Sensations and Flavors Based on Tongue's Electrical and Thermal Stimulation” and It has impressed me. In my opinion, the developed work and field of your research are very interesting as the electrical and thermal simulation of taste sensations has a wide range of potential applications. Moreover, the paper is well written and organized and the conclusions are supported by the experimental and results.

Nevertheless, I have detected some minor errors that should be corrected prior publication. Most of them are directly linked to typewriting so, please, carefully review the full text. Please let me list them point by point:

1.       Introduction, line 25, write “senses” instead of “sense”.

2.       Introduction, line 29, there is an innecesary double space at the end of the sentence.

3.       Introduction, lines 33 and 100, write “researchers” instead of “re-searchers”.

4.       Introduction, line 38, the same with the word “dis-liked”.

5.       Introduction, line 87, the same with the word “ap-plication”.

6.       Figure 1, line, 96, picture (b) is cut on its right. Please correct.

7.       Experimental procedure and result, line 146, maybe it would be better to say men and women.

8.       Figure 4, line 195, “Temperature  (°C)” is enough for text in x-axes.

In my point of view, the manuscript would be suitable for publication after reviewing the above-mentioned aspects.

Finally, I would like to congratulate the authors!

Author Response

The authors response has been attached 

Round 2

Reviewer 1 Report

After carefully evaluating the responses from the author, I am still not convinced about the design of the study. I am listing my responses with respect to the author’s response.

1.     The revised manuscript still doesn’t reciprocate this logic in an easily understandable way. Apart from the lack in logic flow, the design of the study is still questionable. Here’s my understanding: The hedonic scale was restructured between 0 & 1 for the purpose of the study. When stimulated to a flavor, the participants first choose a number (0 or 1) based on if they taste anything at all or not. Then where is the result for that study? i.e., a table about how many of the participants could taste the flavors (both basic and mixed) would give a better understanding if the electrodes are generating the flavors correctly.

2.     Then the author responded that the participants were asked to scale the sensation based on like or dislike. How many participants did you try this with? For example, did you conduct this study for basic taste such as ‘sweet’ for 20 participants and then claimed that more thermal simulation decrease/increase the sweetness? And since taste of sweetness varies between people to people (a person may ‘moderately dislike’ overly sweet taste whereas other person may just have ‘moderate sensation’. The point I am trying to make is that how does the like and dislike towards a taste tell you if you successfully generated a flavor?

3.     When you move to the mixed flavor, things become more complicated since now you have a few participants who can taste it and few who can’t. What was number of participants who could taste something for each mixed flavor?   Did you try the sensation study with the participants who can’t taste the mixed flavor in the first place? If you did, and they can’t, it needs to be included in the study and addressed why.

4.     At one point, the authors responded that the participants were asked to imagine the different taste and then asked to describe the taste they were filling on their tongue. According to the author “most of the participants” could taste bitter taste while asked to imagine sour. “Most of” is very subjective specially when we don’t know if all the 23 participants participated in this study or only the few ones who could taste the sensation at the first place.

5.     In the edited manuscript, the author wrote “some participants were unable to distinguish between different flavors while taking part in the experiment. Though, when they were given a list of various tastes and flavors, the majority of them were able to specify the exact flavor”. You only had 23 participants and some of them can’t distinguish between the flavors at all. In this case, shouldn’t you move forward without these participants? Looking at a chart of flavors in my opinion would affect a participant’s perception of the flavor more than, detecting a flavor just based on sensation.

For this manuscript I would recommend the authors to re-design the experiments. First off start with more participants and screen them based on if they could sense the basic flavors at all. The ones who can then can be asked to distinguish between the flavors on their own. The ones who can distinguish this flavor should then be asked to scale the sensation based on like or dislike at the same time should be asked questions like ‘if this is more sweet or less sweet or moderately sweet?” This will give the author an idea about the sensation scale of the participants and then they can be introduced to mixed flavors to get a more concrete proof that the simulation for digitally mixed flavor is successful. I would also suggest the authors to be clearer and more precise about presenting their data. It would be very helpful to see how many participants between x number of participants could sense sweet taste and then compare against how many of them could sense sweet-sour taste.

Reviewer 2 Report

The Authors have performed the changes requested by the Reviewers. The symbol for micro is with the Greek letter. There was somewhere in the text "uA" (microAmps), which is wrong. However, this was not found in the last version of the manuscript. The paper can now be accepted.